# Inclusive mental health support for international students: Unveiling delivery components in higher education

Halis Sakız[1,2]  and Marty Jencius[3] 

[1]Gerald H. Read Center for International and Intercultural Education, Kent State University, Kent, OH, USA; [2]Department of Educational Sciences, Faculty of Letters, Mardin Artuklu University, Mardin, Turkey and [3]Counselor Education and Supervision Program, Kent State University, Kent, OH, USA

## Research Article

mental health; inclusiveness; delivery components; international students; higher education

**Corresponding author:**
Halis Sakız;
Email: halissakiz@artuklu.edu.tr

## Abstract

This study examines the delivery components of inclusive mental health services in higher education, centering on international university students. Through interviews with 32 participants at a state university in the United States, including students, counseling staff, and faculty, six key themes emerged: mental health professionals' multicultural self-awareness, focus on reparative services, mainstream mental health theories and approaches, professionals' cultural background, faculty involvement and physical space and confidentiality. These findings underscore the importance of training for professionals, expansive mental health offerings, incorporation of diverse approaches, confidentiality, active faculty participation and suitable physical environments. By addressing these components, universities can enhance the quality of mental health support for international student populations, promoting their overall well-being and academic success.

## Impact statement

By delving into the discussion about mental health services within higher education, with a specific focus on international university students, this research unearths vital insights that extend beyond academic discourse. In an increasingly globalized world, where students from diverse backgrounds converge in pursuit of knowledge and mental health concerns continue to affect students' academic and personal experiences, this research highlights the challenges faced by international students and unveils the essential components for fostering effective and inclusive support systems. The identified pillars of successful mental health delivery, including culturally sensitive approaches, comprehensive services and faculty involvement, present considerations for institutions. However, the transferability of these insights must be approached with an awareness of diverse institutional, cultural and contextual factors.

This research contributes to ongoing dialogs on multicultural competency within counseling, offering professionals insights to navigate the complexities of diverse student populations. Rather than providing prescriptive solutions, the study encourages a reflective approach to mental health services, acknowledging the need for tailored strategies that align with the unique contexts of different institutions. While the immediate impact of this research may be most pronounced within the scope of the studied university, the broader conversation it initiates has the potential to influence policy formulation and institutional practices that affect students' overall well-being. By promoting inclusivity, cultural understanding and innovative service delivery, this study offers a catalyst for positive change. It underscores the importance of institutions proactively addressing mental health challenges and investing in comprehensive strategies that accommodate the needs of international students while fostering an inclusive environment for all learners.

## Introduction

In recent years, there has been a global trend of increasing international student enrollment in higher education institutions (Yao et al., 2019). For example, in 2020, international students accounted for 5.5% of total enrollments in the United States and 23.7% in Canada (Institute of International Education, 2020). Although the increase in the United States has plateaued since 2017, likely due to the presidential policies and the COVID-19 pandemic, the United States continues to enroll over a million international students each year (Buckner et al., 2022).

Studying abroad can offer opportunities for personal growth and development. However, it can also present challenges for some students, such as adjusting to a new culture, academic stress (Lipson et al., 2019) and cultural and linguistic stereotypes and barriers (Dryden and Dovchin,

2022). Therefore, mental health services are crucial for supporting students' well-being and success, especially for those with international backgrounds (Forbes-Mewett and Sawyer, 2019).

Despite the recognized importance of mental health services, many services in higher education institutions may not be designed and delivered to address the needs and experiences of international students, creating barriers and resulting in inadequate support for international students (Koo and Nyunt, 2020; Zhai and Du, 2020). Students who perceive mental health services as ineffective or lacking in cultural sensitivity may be less likely to seek help when needed, leading to untreated mental health issues with long-term consequences for the student's academic and personal life (Lee et al., 2014). Moreover, a lack of cultural sensitivity in mental health services can exacerbate cultural barriers and misunderstandings, leading to miscommunication and ineffective interventions (Ching et al., 2017). Therefore, this paper aims to examine the delivery components of higher education mental health services for international students from an inclusive perspective.

### An inclusive approach to mental health

An inclusive approach to mental health recognizes and respects the diversity of clients and aims to provide equitable and accessible mental health services for all individuals, including international students (Sakız and Jencius, 2015; Zunker, 2016; Kourkoutas et al., 2017). This framework emphasizes the importance of considering multiple factors influencing an individual's well-being, including social, cultural, biological and psychological factors (Hick et al., 2009). There are essential principles for implementing an inclusive mental health approach in higher education.

An inclusive approach to mental health acknowledges that individuals are part of complex systems that interact and influence each other (Hick et al., 2009). This is in line with a biopsychosocial model that recognizes that biological, psychological and social factors interact to influence an individual's well-being (Bronfenbrenner, 2005). An inclusive approach considers the interplay between these factors to comprehensively understand the individual's needs and provide effective support (Farrell and Venables, 2009; Sameroff, 2009).

An inclusive mental health approach recognizes that mental health is influenced by various systemic factors, including cultural, social and economic factors (Weare, 2018). Therefore, inclusive professionals consider the impact of these systemic factors on an individual's mental health and well-being to provide effective support (Farrell and Venables, 2009). For international students, this means that their experiences in higher education are shaped not only by their academic and personal experiences but also by the social, political, historical and environmental factors that have influenced their lives (Ozturgut and Murphy, 2009).

An inclusive approach to mental health adopts a preventive and developmental perspective, emphasizing the importance of early interventions and promoting the development of skills and resources that can support an individual's well-being (McLaughlin and Boettcher, 2009; Aival-Naveh et al., 2019). Relying solely on a medical understanding limits the scope of assessment and ignores the impact of various factors on an individual's mental health (McLaughlin and Boettcher, 2009). An inclusive approach recognizes the potential for growth and change and supports the individual's development toward their goals (Pedersen and Ivey, 1993).

To provide effective support to international students, inclusive mental health services consider these broader aspects of student well-being and success and offer a range of services that address the needs and experiences of international students (Kim et al., 2019). Furthermore, an inclusive approach utilizes techniques and approaches that are versatile and culturally sensitive (Weare and Nind, 2011) and that consider the cultural, historical and environmental experiences of all students, including those with an international background (Chavajay, 2013).

### Delivering mental health services for international students

Mental health services in higher education play a crucial role in promoting students' well-being and academic success (Williams et al., 2018; Osborn et al., 2022). Effective delivery of these services is essential to ensure that international students receive the necessary support they need to thrive in a new academic and cultural environment (Minutillo et al., 2020). Conversely, poor delivery of these services can negatively affect international students on campus, hindering their mental health and well-being (Shen et al., 2017).

There are critical components that play an essential role in the delivery of mental health services for international students in higher education. First, accessible mental health services are crucial to ensuring that students receive the necessary support when they need it (Osborn et al., 2022). It is imperative that mental health services are easily accessible to all students, regardless of their background or needs. Additionally, mental health services should be advertised consistently and clearly across campus, and staff should be trained to identify and respond to the needs of international students (Ozturgut and Murphy, 2009). Furthermore, mental health professionals should be available for appointments during convenient hours, and services should be offered at multiple locations throughout the campus. Also, inclusive mental health services require collaboration between mental health professionals, academic staff and other support services on campus. Collaboration ensures that students receive comprehensive and coordinated support services, which can promote academic success and overall well-being (Pedersen and Ivey, 1993).

Cultural sensitivity is a vital delivery component of mental health services for international students (Wen et al., 2022). International students may have diverse cultural backgrounds, beliefs and values and may face challenges related to their cultural identity (Keum et al., 2022). Professionals need to be aware of the cultural differences between them and their clients and the challenges that international students face when adapting to a new academic and cultural environment (Sue and Sue, 2016; Shen et al., 2017).

Providing a variety of services is another crucial delivery component of mental health service for students (Constantine and Ladany, 2000). International students come from diverse backgrounds and may have some needs and concerns, which may require various types of counseling services. Therefore, it is essential to offer a range of mental health services, such as individual counseling, group counseling, career guidance and family counseling, to meet the diverse needs of international students (Sue and Sue, 2016).

### Research gap, significance and question

Despite the growing number of international students attending higher education institutions worldwide (De Wit and Altbach, 2021), which reached 6.1 million in 2019 (Tsiligkiris and Ilieva, 2022), there is still a significant gap in the research on the delivery components of inclusive mental health services which can include this population. While some literature underlines international students' mental health needs (Sue and Sue, 2016; Li et al., 2017;

Anandavalli et al., 2020; House et al., 2020; Osborn et al., 2022), there is a lack of research on how to provide effective and inclusive mental health services to all students, including those with international background.

Research on the delivery components of inclusive mental health services for international students is significant for several reasons. First, international students comprise almost 3% of the student population in higher education institutions worldwide (UNESCO, 2022). Therefore, it is critical to ensure that mental health services are inclusive and accessible to promote students' well-being and academic success. Second, research in this area can provide insights into international students' needs and experiences, which can inform the development of mental health services tailored to their needs (Sue and Sue, 2016). For example, international students may have specific needs related to acculturation and cultural adjustment, and understanding these needs can help mental health professionals develop interventions that promote coping strategies and resilience among international students. Third, research in this area can address the underrepresentation of international students in the mental health literature (Hamamura and Laird, 2014; Nam et al., 2023). Therefore, answers to the following research question were sought: What are the essential delivery components of inclusive mental health services for international university students?

## Methodology

### Research design

This is a qualitative study aimed at understanding participants' subjective experiences and perspectives about the delivery of mental health services for international students with an inclusive approach (Lichtman, 2014). The case study approach also allows for data collection from multiple and different stakeholders, providing a more comprehensive understanding of the topic and increasing the validity of the findings (Yin, 2009).

### Participants

The study involved 32 participants, including students and staff, from a state university in the United States. Participants had a mean age of 36.7 years. The selection of all participants was guided by a rationale to ensure a well-rounded understanding of mental health services. A hybrid of purposive and snowball sampling was utilized for the sampling strategy (Flick, 2007). Purposive sampling was employed to select participants with direct knowledge and experiences of the mental health services provided by the university. Snowball sampling was used to identify additional participants who could provide further insights into the experiences and perceptions of mental health services provided to international students.

The identification and recruitment of the participants in each group were guided by their roles and relevance to the study objectives. For instance, managers overseeing counseling and psychological services units, faculty from pertinent departments, staff directly involved in mental health services and managers of sociocultural and inclusive support units were purposefully selected based on their expertise. Initial participants were approached based on their key roles within these units, ensuring their insights could provide a foundational understanding of the mental health support landscape. The sample size was determined through the iterative process of thematic saturation, where new participants were included until no novel themes or insights emerged, ensuring a comprehensive exploration of diverse perspectives within a manageable scope. The sample size was further guided by the principle of adequacy, aiming for a balance between depth of insight and practical research constraints.

Thirteen participants were international students, including six undergraduate and seven graduate students. Additionally, four participants were American students, comprising two undergraduate and two graduate students. All students had firsthand experience with mental health services, having received these services. Characteristics of the students are provided in Table 1.

The study recognized the importance of capturing the perspectives of both international and American students engaged in higher education mental health services. While 13 of the 17 student participants were international, representing a crucial demographic, the experiences of the remaining American students should be acknowledged. They were included intentionally to ensure a comprehensive understanding of mental health challenges within the university context.

This study included 15 staff participants who played key roles in various units and departments within the university, offering a comprehensive view of mental health services. The staff were purposefully selected based on their roles and expertise, ensuring a rational and insightful representation of the university's support infrastructure. The diverse composition of staff participants included (i) three managers of counseling and psychological services units, who held managerial positions overseeing the delivery of counseling and psychological services to students, providing crucial insights into the administrative aspects of mental health support, (ii) five counseling and psychological services staff, who comprised practitioners directly involved in providing counseling and psychological services and contributed frontline perspectives on the daily challenges and interactions within the mental health support framework, (iii) three managers who were responsible for managing units that provided a spectrum of support services, including sociocultural, career, academic, accessibility and inclusive support, and added a broader context to the study by encompassing a range of student services and (iv) four faculty members from counseling (n = 2), psychology (n = 1) and sociology (n = 1) departments who ensured academic perspectives on mental health services and brought valuable insights into the intersection of mental health support and academic disciplines. Each participant, regardless of their role, provided helpful information during the study, contributing to the richness of the data and the depth of analysis, reflecting a diversity of experience. However, specific demographic details of the staff, including cultural identity and intercultural experiences, were not provided to ensure that any information that could reveal the identity of the staff members is confidential.

### Study context

This research constitutes a single-site collective case study situated at a university in the United States. The university's commitment to inclusion, equity and diversity is enshrined in its policy framework. This commitment is expected to permeate the provision of mental health services, emphasizing a student-centered approach that respects and responds to individual, cultural and linguistic diversities. Therefore, the university's mental health services are designed to recognize and respect the cultural, linguistic and individual variations among its students.

The university's mental health services encompass a range of offerings, including individual counseling, group therapy sessions

**Table 1.** Characteristics of participating students

| Participant | Student | Gender | Age | Academic level | Department | Cultural background |
|---|---|---|---|---|---|---|
| 1 | International | Male | 28 | Graduate | Engineering | Chinese |
| 2 | International | Female | 32 | Graduate | Education | Indian |
| 3 | International | Male | 33 | Graduate | Business | Middle Eastern |
| 4 | International | Lesbian | 29 | Graduate | Literature | European |
| 5 | International | Male | 25 | Graduate | Education | South American |
| 6 | International | Female | 24 | Graduate | Theater | African |
| 7 | International | Male | 36 | Graduate | Physics | Asian |
| 8 | International | Female | 22 | Undergraduate | Nursing | Southeast Asian |
| 9 | International | Male | 23 | Undergraduate | Business | Middle Eastern |
| 10 | International | Female | 20 | Undergraduate | Education | South American |
| 11 | International | Male | 26 | Undergraduate | Political science | Indian |
| 12 | International | Female | 19 | Undergraduate | Arts | Asian |
| 13 | International | Bisexual | 21 | Undergraduate | Media/journalism | European |
| 14 | American | Male | 33 | Graduate | Chemistry | African American |
| 15 | American | Female | 26 | Graduate | Communication | Caucasian |
| 16 | American | Gay | 24 | Undergraduate | Engineering | Hispanic |
| 17 | American | Female | 22 | Undergraduate | Education | Caucasian |

and educational events. These services are structured to cater to the diverse needs of students, acknowledging that mental health support is not a one-size-fits-all endeavor. Moreover, the university actively fosters a welcoming environment for international students through on-campus cultural services and programs. These initiatives are integral to helping international students feel embraced and celebrated for their cultural richness. By organizing various cultural events, workshops and programs, the university enriches the overall student experience and contributes to a supportive atmosphere.

At university, students, both international and domestic, can access mental health services through multiple channels. The university employs a confidential and user-friendly appointment system, facilitating ease of access for students in need. To align mental health services with the needs of international students, the university conducts needs assessments. These assessments, often facilitated through intake interviews, enable mental health professionals to tailor services to individual requirements. The negotiation of services involves a collaborative approach between students and mental health professionals, ensuring a client-centered and culturally sensitive approach.

Mental health services at the university follow a structured approach. The services are managed by distinct units, including psychological services and the counseling center. These units collaboratively engage in negotiations to tailor services that meet the diverse needs of the student body. This collaborative effort ensures a comprehensive approach to mental health support.

### Data collection instruments

The data collection instruments used in this research were semi-structured interviews designed specifically for each participant group. The questions were developed after reviewing related literature (Ching et al., 2017; Lipson et al., 2019; Keum et al., 2022) and

considering the study objectives. Four pilot interviews were conducted to refine and validate data collection instruments (Flick, 2007; Cresswell, 2009). In each interview, the questions were carefully assessed for cultural sensitivity by seeking feedback from individuals with diverse cultural backgrounds. The pilot participants included two international students, one American student and one faculty member, representing a variety of cultural contexts and ensuring that the questions resonated appropriately across different perspectives. This iterative process allowed to make adjustments and modifications based on insights obtained from the pilot studies.

The determination of cultural sensitivity involved direct engagement with individuals from distinct cultural backgrounds, who provided feedback on the clarity, relevance and appropriateness of our questions. Their input was instrumental in shaping the final version of the schedules. No incentives were provided for participation in the pilot studies, ensuring that feedback was genuinely driven by the participants' willingness to contribute to the improvement of the study.

The semi-structured interviews were crafted to cater to the distinct perspectives of each participant group. The development process involved a robust and iterative approach to ensure rigor and relevance. The decision to employ different interview schedules for each participant group was underpinned by the recognition of their unique roles, experiences and contributions to the university's mental health support landscape. Tailoring the questions allowed for a focused exploration of the specific challenges, needs and insights pertinent to each group. This approach aimed to elicit in-depth responses attuned to the diverse contexts in which participants operated.

The interviews covered a broad spectrum of topics, delving into participants' perspectives on the overall effectiveness and inclusivity of mental health services. Participants shared insights on cultural sensitivity, specific challenges faced, potential areas for

improvement, the intersection between mental health support and academic environments, collaboration, the nature of mental health considerations within a university setting and complexities surrounding mental health services for international and American students, faculty members, unit managers and practitioners.

The number of the questions for each participant group varied based on the content and purpose of the interview (e.g., 'management of the services', 'faculty involvement in services'). There were 12 questions for international students (e.g., "How would you describe your overall experience with the current mental health services offered by the university?", "Do you feel the current services are inclusive and sensitive to diverse cultural backgrounds?"), eight questions for American students (e.g., "Can you share your perspective on the inclusivity of mental health services, particularly in catering to the needs of the students?", "How do you perceive the cultural inclusiveness of the mental health programs available to students?"), 10 questions for faculty members (e.g., "As an academic, how do you perceive the intersection between mental health support and the academic environment?", "What role do you envision academic departments playing in fostering a mentally healthy campus?"), nine questions for unit managers (e.g., "How do you currently assess the effectiveness of your unit's mental health services for students?", "Could you speak to the measures taken to ensure the inclusiveness of mental health initiatives within your unit?") and 10 questions for practitioners (e.g., "Could you elaborate on the daily challenges you encounter while providing counseling and psychological services?", "In your role, how do you contribute to fostering an inclusive and supportive environment for diverse student needs?").

### Procedure

Before conducting the study, an institutional review board approval from the university was obtained. All participants were informed of the purpose and procedures of the study, and they provided informed consent before participating in the interviews. Data collection was performed from December 2022 to February 2023 in four phases: (1) a literature review to develop interview schedules, (2) the design and validation of the interview schedules, (3) data collection through in-person (n = 27) or remote (n = 5) interviews with participants and (4) data analysis.

During the interviews, open-ended questions were asked to understand the participants' experiences with higher education mental health services and the delivery components of inclusive services. The interviews lasted approximately 1 hour and were conducted in a comfortable and private setting to facilitate open and honest communication. Detailed notes were taken, and audio recordings were made for the interviews.

### Data analysis

The study analyzed interview data using thematic and inductive analysis to identify patterns and build themes from the data itself. Thematic analysis is a widely used technique for categorizing meaningful themes and patterns (Braun and Clarke, 2006), while inductive analysis allows for a bottom-up approach to data analysis and provides a more grounded understanding of the participants' experiences (Saldaña, 2013). The use of thematic and inductive analysis in this study enabled a comprehensive understanding of the participants' experiences. First, the study aimed to answer the research question from the perspectives of different stakeholders. Thematic analysis allowed for the identification and categorization

of recurring patterns within the data, providing insight into the experiences and perspectives of the participants. Additionally, inductive analysis facilitated a bottom-up approach to data analysis, where themes and patterns were identified based on the data itself rather than on preexisting assumptions or theories.

The data were transcribed, and initial codes and themes were identified through open coding. The themes were then linked together through axial coding, and selective coding was used to synthesize the themes into a meaningful representation of the data (Corbin and Strauss, 2008). These techniques ensured systematic and rigorous data analysis, robustly representing the participants' experiences and perspectives.

During thematic analysis, an inclusive approach was adopted, considering both commonalities and distinctions in the experiences shared by American and international students. It was also recognized that American students might have undergone significant transitions and challenges related to relocation, dislocation or feelings of otherness. These factors, common to both student groups, were crucial in understanding the broader spectrum of mental health issues. Therefore, the analysis aimed to identify overarching themes that resonated across diverse backgrounds, allowing for an exploration of shared challenges and unique concerns. American students' experiences were not excluded; instead, they were integrated into the thematic analysis, acknowledging the potential convergence or divergence of themes across different student groups. This ensured that the voices of all participants, local and international, contributed meaningfully to the rich tapestry of perspectives explored in the study.

### Trustworthiness

Trustworthiness is crucial for ensuring the validity and reliability of qualitative research findings. This study employed several strategies to enhance trustworthiness (Huberman and Miles, 2002). To ensure credibility, the study involved an extended data collection period and analyzed the data in-depth, cross-checking the findings with existing literature. To ensure transferability, the study used a hybrid of purposive and snowball sampling techniques to collect data from diverse participants. To ensure dependability, the study employed a rigorous data analysis process that involved multiple researchers analyzing the data and cross-checking the results. These measures demonstrate the robustness of the findings, ensuring that the results of this study are trustworthy.

### Findings

The study yielded six themes discussed in this section (Table 2). These themes provide insights into the essential delivery components of inclusive mental health services for international students and offer recommendations for improving university counseling services.

### Mental health professionals' multicultural self-awareness

Effective and inclusive mental health support for international students requires professionals to have adequate training in multicultural support. One practitioner emphasized this need by stating, "Professionals need to be equipped with knowledge of cultural differences to provide effective mental health services to international students." However, participants also identified a shortage of culturally knowledgeable and sensitive staff. An international

**Table 2.** Themes and subthemes

| Themes | | Subthemes |
|---|---|---|
| Professionals' multicultural self-awareness | → | • Training<br>• Culturally knowledgeable and sensitive professionals<br>• Students' cultural concerns and assessment<br>• Taboo and severity of problems<br>• Dropout |
| Focus on reparative services | → | • Individual support<br>• Universal approaches |
| Mainstream mental health theories and approaches | → | • Intercultural applicability<br>• Beliefs on mental health<br>• Altering meaning of recovery |
| Professionals' cultural background | → | • Common therapeutic factors<br>• Disclosure<br>• Satisfaction |
| Faculty involvement | → | • Supervision and training<br>• Collaboration and support |
| Physical space and confidentiality | → | • Waiting and counseling/therapy rooms |

student mentioned, "Professionals need to be trained to work with international students to provide culturally responsive care." Another perspective was shared by an American student who emphasized the diversity even within American origins, stating, "Being American-born, I still value having a counselor who understands the diverse cultural backgrounds within our own country. It's important to have someone who recognizes the richness of our various backgrounds and experiences."

To address the need for more culturally responsive mental health services, participants emphasized the importance of tapping into students' cultural concerns, starting with the need for better earlier assessments. A faculty member highlighted this by emphasizing, "Early assessments allow us to recognize the individuality of each student, plan accordingly and provide more effective and inclusive mental health support from the beginning."

Importantly, this research also captures the perspectives of American students, shedding light on their views of mental health support within the university context. An American student shared, "The need for culturally knowledgeable staff is crucial. I, an American student, also benefit from professionals who understand the diversity we bring to the university environment."

There was a view of mental health support seen as a taboo within some cultures. One manager expressed the need for professionals to consider this: "Some students come from cultures where talking about mental health is seen as taboo. Mental health professionals need to be sensitive to this and find ways to make mental health support more culturally acceptable and accessible to them." In line with a taboo view, some students did not tend to seek support until their problems became severe. An international student emphasized the importance of addressing this issue: "Many international students tend to wait until their problems become unmanageable before seeking support. More ways are needed to encourage them to seek help earlier on before problems escalate."

Another issue highlighted was a dropout of mental health support. The absence of a clear treatment plan is not only a factor in client dropout but also relates to multicultural self-awareness. As an international student emphasized, "It is important to have a plan, and I feel like sometimes professionals don't have one." A

well-structured plan is essential for acknowledging diverse cultural perspectives and delivering interventions that align with the unique needs and backgrounds of clients. The lack of such planning may hinder providers in navigating cultural nuances, potentially contributing to dissatisfaction and dropout.

### *Focus on reparative services*

The second theme was the overemphasis on reparative approaches, while developmental and preventive approaches were overlooked. Firstly, there was a preponderance of clinically oriented individual therapy, which was regarded as the default service provided to students. An international student said, "Every time I went to the support center, they recommended individual counseling. It's like they didn't have any other options. While in individual counseling, I felt like the focus was on addressing immediate concerns, rather than delving into aspects of my relationships, and the cultural nuances contributing to my challenges." However, several participants viewed individual counseling as ineffective and inadequate in addressing the multifaceted challenges faced by international students, especially when it was overly clinical and pathologically oriented. As one international student observed, "I felt like my psychologist didn't understand the cultural nuances and just gave me generic advice that didn't really help me. It felt as though they insinuated that I had individual deficiencies, rather than addressing how others relate to me."

Next, participants felt there was a lack of approaches focused on prevention, education, awareness and developmental support. They believed these services were crucial in creating a supportive and inclusive environment for international students. An American student emphasized the importance of preventive approaches, stating, "I wish they had more workshops and events to teach me coping skills and stress management before things got really bad."

### *Mainstream mental health theories and approaches*

Participants reported that mainstream theories and approaches were dominant, while the applicability of these could not be high for some cases. One international student stated, "I feel like the mental health services are tailored toward Western students. I feel like the theories and approaches used by professionals are not inclusive of different cultural perspectives and values." The dominance of mainstream and Western approaches and theories can be less considerate of international students.

Participants' views showed that there were cultural differences in beliefs on mental health. An international student stated that "In my culture, mental health is not something we openly talk about or seek help for. It would be helpful if counselors could consider these differences and not assume that all clients have the same beliefs and attitudes toward mental health." Relatedly, the meaning given to recovery by different cultures could also vary. Another international student expressed that "In my culture, recovery from mental health does not mean that you are problem-free. Also, the emphasis is not just on the individual but also on the family and community. Professionals could take this into account and adopt a more holistic approach to recovery."

### *Professionals' cultural background*

The fourth theme of the study delves into professionals' cultural background. First, participants reported the positive effects of

common therapeutic factors used by an American practitioner, such as unconditional positive regard and empathy. One international student stated, "My counselor's empathy made me feel heard and validated. It helped me open up and trust her." Also, there was a preference for an American counselor because disclosing personal information to someone from their own culture may be seen as shameful for some students. An international student shared, "I feel less comfortable talking about my problems with someone from my own culture. I feel like they would blame me for my struggles."

However, some students were dissatisfied with the practitioners due to a lack of multicultural competency. An international student expressed, "I felt like my therapist didn't understand my cultural background, it was difficult to explain things to them. It made me lose trust in the therapy process." In addition, some participants thought that language barriers could hinder effective communication between professionals and clients. An international student explained, "It's hard to express myself in English, and sometimes, I feel like I'm not being understood." Overall, this theme indicates the importance of having humanistic values and multicultural competency to provide effective mental health services to international students.

### Faculty involvement

To explore the faculty involvement in mental health support delivery, the participants emphasized the need for faculty to be involved more in the delivery and supervision of mental health services as well as the education of professional candidates. According to a faculty, "Having faculty involvement in the training and supervision is crucial for developing multicultural competence in future mental health staff." However, some participants pointed out that faculty involvement is limited, affecting mental health services' overall effectiveness. One manager said, "Faculty involvement in mental health support delivery is limited. However, involvement would enhance the impact of mental health services on students." An international student who emphasized the need for faculty engagement in mental health services stated, "Faculty members can contribute significantly to creating a supportive environment. Their involvement is valuable for students in enhancing the overall effectiveness of mental health support services."

### Physical space and confidentiality

The study also delved into the confidentiality and privacy of mental health support. First, the waiting and therapy rooms did not sufficiently protect the identities of the clients and professionals. Regarding the waiting room, one American student shared, "There were other students in the waiting room who were not there for support, and they could easily tell who was waiting for mental health services. I felt uncomfortable and exposed." An international student added, "The waiting room was open, and people could see you through the window. I think it is important to have a private and secure waiting room."

Similarly, an international student stated, "The therapy room was not soundproof, and I could hear other people talking outside. It made it difficult for me to focus on my issues and felt like my privacy was being invaded." An American student added, "The counseling room was too small, and it felt cramped. The therapist was also seated very close to me, and it made me uncomfortable to discuss personal issues." These findings highlight the need for institutions to prioritize the privacy and confidentiality of mental health support.

## Discussion

The findings of this study shed light on the delivery components of inclusive mental health services in higher education. Through interviews, the study identified several key themes that impact providing mental health services for international students. The discussion explores these themes in detail, highlights the implications of these findings and suggests recommendations for delivering inclusive mental health services in higher education.

The study's first finding emphasizes the significance of mental health professionals having an awareness of cultural differences to provide effective support for students. This finding aligns with prior research emphasizing cultural competence's significance in training (Sue and Sue, 2016; Osborn et al., 2022). The study's findings highlight various areas in which mental health services for students can be enhanced to align with inclusive principles. For example, it is crucial for professionals to have knowledge and comprehension of their clients' cultural backgrounds to deliver effective mental health support (Osborn et al., 2022). As Hannon and Vereen (2016) note, cultural competence is essential to mental health support. Inclusive mental health values and recognizes diversity, promoting accessible and equitable services for all individuals, irrespective of their cultural background or identity.

Another finding was international students' dropout of mental health support, which was attributed to language barriers, students' perceptions of being misunderstood and a lack of treatment planning and structure. Previous research highlighted the reasons behind the discontinuation of mental health services among international students, which includes dissatisfaction with the quality of services, inadequate cultural sensitivity and lack of perceived benefit (Kronholz, 2014). The study further emphasizes that international students are likelier to drop out of services when the interventions do not meet their cultural needs and preferences (Kim and Zane, 2016).

Language barriers can impede the effectiveness of mental health services for international students, as students may have limited proficiency in English (Wen et al., 2022). A lack of proficiency in the language can make it difficult for international students to express their concerns and feelings, leading to miscommunication with professionals. Moreover, due to cultural differences, international students may feel misunderstood by the professionals, leading to a lack of trust in the process and resulting in dropping out of mental health services (Williams et al., 2018). Culturally sensitive and inclusive mental health support are essential to address these challenges and ensure negative fixed perceptions and judgments based on language and culture do not lead to feelings of incomprehensibility and insufficiency (Dovchin, 2020).

In addition, the lack of treatment planning and structure in mental health support may contribute to international students' dropping out of services. Students from different cultural backgrounds may have different expectations of mental health services and may not understand the structure and purpose of sessions in each context (Wen et al., 2022). This can lead to a perceived lack of benefit and low engagement with mental health services.

The findings of this study underscore the importance of addressing cultural concerns in mental health services for international students. First, early assessment to identify international students'

specific needs and cultural background is crucial to understanding their needs. As Arthur and Popadiuk (2010) pointed out, early assessment can enhance the effectiveness of mental health services and ensure that appropriate interventions are implemented. Cultural differences in attitudes toward mental health may require an additional assessment to understand and overcome potential barriers to treatment. Secondly, the results highlight the stigma surrounding mental health in certain cultures, which leads to the view of mental health support as taboo. International students may hesitate to seek mental health support due to fear of judgment, shame or social stigma associated with mental health issues until problems become severe. Vogel et al. (2017) noted that cultural values and beliefs could influence perceptions and attitudes toward mental health, making it challenging to encourage international students to seek help.

The study's findings indicate a high focus on reparative services and approaches while ignoring developmental and preventive approaches. The professionals mainly provided individual mental health services, which can be individualistic and overlook clients' social and cultural context. Focusing solely on reparative services and approaches in mental health can limit the potential for growth and development in individuals, including international students in higher education (McLaughlin and Boettcher, 2009). Reparative approaches are often reactive and focused on addressing current problems or symptoms rather than taking a preventative or developmental approach that addresses the underlying causes of those problems (Constantine and Ladany, 2000). However, inclusive mental health recognizes the importance of holistic, proactive and contextual approaches that consider the broader social and cultural contexts that shape clients' experiences.

The findings highlight the challenges associated with mental health services. An exclusive focus on reparative approaches may not be effective for all international students, as they may come from diverse cultural backgrounds that may require different approaches to mental health. For instance, some cultures may prioritize collective well-being over individual well-being and may benefit more from preventative and developmental approaches to mental health (Sue and Sue, 2016). Therefore, inclusive mental health services consider cultural differences and provide a range of services tailored to international students' diverse needs and backgrounds (Sakız and Jencius, 2015).

Studies suggest that universal, preventive and educational programs that focus on enhancing coping skills and promoting mental wellness may effectively address students' needs (Weare, 2018). Therefore, professionals need to develop and implement prevention and education programs that are culturally sensitive and responsive to student needs more holistically and inclusively (Letourneau, 2016), focusing on promoting mental wellness rather than just the treatment of mental illness. Moreover, inclusive mental health services can provide a range of group services tailored to international students' needs (Hick et al., 2009).

One major concern that has been identified is the applicability of mainstream mental health theories and approaches to international students, which may not consider the cultural and contextual factors that influence their experiences. Sue and Sue (2016) noted that traditional Western theories might not be suitable for individuals from different cultural backgrounds, as these theories may not be relevant to their experiences and may not capture the impact of cultural differences. This concern was echoed by some participants in this study who reported feeling that the services did not consider their cultural and personal backgrounds.

Some cultures view mental health differently from the Western perspective. For instance, seeking mental health support may be seen as a last resort or a sign of weakness in cultures that view mental health from a clinical and crisis perspective. Conversely, some cultures may view mental health as an integral part of a life and wellness routine, where mental health services are seen as a form of self-care (Mowbray et al., 2006; Forbes-Mewett and Sawyer, 2019). These cultural differences highlight the importance of inclusive mental health services that are tailored to the needs and backgrounds of international students (Mori, 2000; Sakız and Jencius, 2024).

In addition, recovery can have different meanings for people from different cultures, and the definition of recovery needs to be tailored to everyone's cultural background (Aival-Naveh et al., 2019). The emerging focus on 'recovery' in the design of services and their outcomes fundamentally alters long-term options for people with serious mental illnesses (Farrell and Venables, 2009). A recovery-oriented approach involves encouraging individuals to take different risks, leading to various successes in their lives and goals. The impact of this recovery philosophy is spreading into education, enabling more people with mental problems to continue pursuing educational goals and achieving success. Colleges and universities should prepare for this possibility and embrace it (Mowbray et al., 2006).

Clients' satisfaction with an American mental health professional is a complex issue influenced by various factors. One finding is the positive effects of common therapeutic factors used by international students, such as unconditional positive regard and empathy. Pedersen and Ivey (1993) noted that a caring relationship that includes empathy, positive regard and genuineness can facilitate the counseling process regardless of the counselor's cultural background. Indeed, findings indicate that students may prefer an American professional, as disclosing personal information to a culturally familiar professionals could be perceived as shameful or weak. This is particularly true for individuals from collectivist cultures (Pedersen, 2000). However, professionals who lack multicultural competence may unintentionally perpetuate cultural biases and stereotypes, leading to a breakdown in the therapeutic relationship (Forbes-Mewett and Sawyer, 2019).

Faculty involvement in mental health support is a crucial aspect of providing quality services to international students on college campuses. It is essential for faculty to provide training and supervision in the process of providing effective support to international students. Faculty can offer guidance on cultural competence and help candidates develop the necessary skills to work with international students. Faculty involvement in supervision and training positively impacts candidates' ability to work with diverse populations (Kress and Protivnak, 2009). Collaboration can involve joint training and program development, as well as joint research and evaluation of services. Faculty involvement can ensure that mental health programs align with international students' needs. Faculty members can also serve as role models and advocates for inclusive mental health, which can help to reduce stigma and promote a culture of inclusion on campus.

The final insight from the study underscores the imperative of enhancing mental health support to guarantee confidentiality for all students, irrespective of their background (Koo and Nyunt, 2020). Participants emphasized that both waiting and therapy rooms lacked sufficient safeguards to protect the identities of clients and professionals alike. This poses a considerable risk for students, whether international or American, who are seeking

mental health support. The inadequacy in privacy measures can create an uncomfortable environment, hindering students from openly discussing sensitive or personal matters. Consequently, this may lead to diminished satisfaction with services and potentially discourage students from seeking support. Recognizing the connection between privacy and inclusiveness, it is paramount for mental health service units to prioritize and implement robust measures ensuring confidentiality. This commitment can foster an environment where all students can avail themselves of effective and inclusive mental health services (Yakunina and Weigold, 2011).

## Limitations and recommendations

This study has some limitations that must be acknowledged. First, the study was conducted in a single-state university in the United States, and the findings may not be reflective of those attending other higher education institutions, countries and cultures. Second, the study discovered and focused on a specific set of delivery components, potentially overlooking other relevant factors that influence the effectiveness of counseling services, such as institutional policies, societal attitudes toward mental health and broader cultural influences. It is essential to acknowledge that other unexplored components may also play a role in supporting international students' mental health. Third, the absence of academic perspectives from the departments where the participant students were enrolled is a recognized limitation of this research. Future inquiries should aim to bridge this gap by incorporating insights from academic staff. Finally, staff members' specific demographic details were not provided to ensure confidentiality. However, this could also provide context to make more meaning out of the study findings.

Based on the findings, the following recommendations are proposed. First, higher education institutions should prioritize multicultural and self-awareness training for professionals providing mental health services. Second, mental health services should not only focus on reparative services but also include proactive services that promote the well-being of international students. Third, higher education institutions should promote the integration of diverse mental health approaches and theories to ensure inclusivity. Fourth, physical space and confidentiality must be ensured to provide a safe and welcoming environment for mental health support. Finally, faculty members should be encouraged to be more involved in mental health services for international students, especially in promoting awareness and support.

**Open peer review.** To view the open peer review materials for this article, please visit http://doi.org/10.1017/gmh.2024.1.

**Data availability statement.** The data are available upon request from the authors.

**Acknowledgements.** We thank all participants for their contribution to the study.

**Author contribution.** Conceptualization: H.S. and M.J.; data analysis: H.S. and M.J.; data collection: H.S.; methodology: H.S.; writing—original draft preparation: H.S.; writing—review and editing: M.J. All authors have read and agreed to the published version of the manuscript and agree to be accountable for all aspects of the work in ensuring that questions related to the accuracy or integrity of any part of the work are appropriately investigated and resolved.

**Financial support.** This research was funded by the Scientific and Technological Research Council of Turkey (TUBITAK) within the scope of 2219 Post-Doctoral Research Scholarship Program.

**Competing interest.** The authors declare no competing interest exists.

**Ethics statement.** The study was conducted in accordance with the approved guidelines and regulations from the Research Ethics Committee of Mardin Artuklu University (protocol code 34233153-050.04.04, March, 20, 2023). Informed consent was obtained from all participants involved in the study.

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
