## [Reviewer Report]

08.12.2023

Dear Professor Gary Belkin, PhD

We are happy to submit our article entitled “Inclusive Mental Health Support for International Students: Unveiling Delivery Components in Higher Education” to Cambridge Prisms: Global Mental Health.

The study investigates the delivery components of higher education mental health services for international students through an inclusive approach. By conducting this study, we believe that we can better understand the needs of international students and develop more inclusive and effective mental health services at higher education. This will not only benefit international students but also contribute to the development of a more diverse and inclusive higher education system. This case study was conducted through interviews with 32 participants from a state university in the United States. We believe that many other countries going through similar procedures in terms of internationalization can benefit from findings of this study. 

We hope that your editorship finds the manuscript valuable for review. We agree that the manuscript has not been published nor submitted for publication elsewhere. Both authors in this research declare no potential conflicts of interest with respect to the research, authorship, and/or publication of this article. 

With kind regards,

Halis Sakiz, PhD

Associate Professor

Corresponding author

---

## [Reviewer Report]

I think that your paper addresses an important issue for university providers. The paper was well written and appropriately located in the research literature on the topic. I hope that my comments will be helpful in revising the paper for publication.

Impact of the study

The authors claim that this “investigation’s impact is not confined to a single university or locale; it resonates with academic institutions globally” (p. 1, line 22); this research reverberates across boundaries, bringing forth a heightened awareness of the importance of tailored mental health services in higher education… steering universities worldwide toward a future where mental health support is as diverse and dynamic as the student body it serves” (p1, lines 48-58). These claims are at odds with the limitations acknowledged in the last paragraph: the research is a single site case study in a state university in the USA; the study discovered and focused on a specific set of delivery components, potentially overlooking other relevant factors which may influence the effectiveness of counselling service.” While the issue of international students’ mental health may resonate with academic institutions globally, the claims in the impact statement seem hubristic, given the scope and scale of the investigation reported in this paper.

Conceptual conflict

There is a conceptual conflict between a recognition of diversity of cultural background and unique challenges related to the cultural identity of international students. They are characterized as having diverse backgrounds with “unique needs and concerns”, “unique needs of international students”, “unique needs and experiences” (p. 6, lines 6, 33, 49 respectively.) Even with this emphasis, there is a conceptual blurring in which the unique becomes homogenized in the collective of “international students”. In some respects, the emphasis on the “unique” needs of international students may contribute to one of the practices criticized at the research site – what is seen as an “overemphasis on reparative services and approaches”. The “unique” focus is also seemingly at odds with one of the reported findings: a desire attributed to participants for “universal approaches focused on prevention, education, awareness, and developmental support”.

Research design

The study is a single site collective case study at a state university in the USA, intended to understand participants' subjective experiences and perspectives about the delivery of mental health services for international students with an inclusive approach. There is no site-specific contextual detail supplied for this study. For example, how is the inclusive approach to mental health service operationalized at the research site? What services are provided, how do students (international or otherwise) access those services, how are the needs of client students determined, how are the relevant services negotiated and how are they delivered? Without some contextual information of this kind, there is no evidence to support the claim that the practices engaged with at the site are designed to be inclusive and no contextual framework in which to locate the findings of the study. What is this study a case of? Is it an evaluative case study of the ways in which inclusive mental health services are provided at the site? If so, then the contextual detail outlined above is essential to understanding the case and the recommendations which arise from it.

One might have expected that the perspectives of international students might be a substantial element in this study, but that is not evident. Of the 32 participants, 17 were students: 13 international (6 undergrad, 7 grad); 4 local (2 undergrad, 2 grad). They were enrolled across twelve different departments in the university, which should indicate a diversity of academic experience, although this aspect apparently is not considered in the study. No information was provided on cultural demographics of student participants which might have offered some insight into how cultural diversity and unique needs and backgrounds was constituted in this study. How is a reader to construct the cultural background of the local students in contrast to the international students? Rural? African American? First Nations? Transgender? Christian fundamentalist? This group has also been homogenized as “local”. The researchers recruited participants through purposive and snowball sampling, so it seems reasonable to assume that the student participants had direct experience of the mental health services provided by the university.

The 15 staff participants seem to have been drawn almost exclusively from various student support services: three managers of units providing counseling and psychological services; five staff members providing counseling and psychological services; three managers of units providing socio-cultural, career, academic, accessibility, and inclusive support; and four faculty members from the counseling, psychology, and sociology departments. No academic staff who might have offered a different perspective on the needs of international students were included from departments in which the students were enrolled. No information about the cultural identity or intercultural experience of staff members was offered. The only demographic information was that the mean age of participants was 36.7 years, which given that participants’ status ranged from undergraduate students to relatively senior managers seems unhelpful.

Data Collection

Considerable effort appears to have been put into developing valid interview schedules. Schedules initially developed from the literature were examined by the researchers and other experts in the field, cross-checked with related literature, tested through pilot interviews, and modified based on feedback from the pilot participants. Different schedules were developed for each participant group, but there is no discussion of the rationale for that decision, or whether there was any consideration of the differences in responses of the different groups. None of the interview schedules, or even interview questions which turned out to be particularly salient have been shared with readers; they have to be deduced from selected reportage in the findings and discussion.

Data were collected through 32 interviews and included detailed notes. Although “audio recordings were made for select interviews” there is no indication of how many of these there were, how they were selected for recording, how data were collected from the other interviews or why it was decided to treat some interviews differently from others.

Presentation of findings

Thematic and inductive analysis using grounded theory practices produced six themes. These were discussed with illustrative quotes from participants. The researchers used open-ended questions during the interviews to understand the participants' experiences with higher education mental health services and the delivery components of inclusive services. Thematic analysis allowed for the identification and categorization of recurring patterns within the data, providing insight into the experiences and perspectives of the participants;” the study aimed to answer the research question from the perspectives of different stakeholders” (p.9, line 9). There is no explanation of how the perspectives of different stakeholders were recognized or addressed. For example, four of the 17 (23.5%) student participants were “local”; did their perspectives and experiences differ in any way from those of the international students? How were these data treated in the thematic analysis? Were they excluded, in which case what was the point in collecting them in the first place? Were they perceived not to differ substantially from the data of the international students, and simply rolled into the thematic analysis? Given that in the USA, many students relocate across the country to access higher education, some apparently “local” students could have experienced dislocation and alienation as a result; mental health issues for “local” minority groups could have been further heightened by the feelings of otherness (Letourneau, 2016). In such circumstances, data from local students may not be substantially different from that of international students. There is simply no way in which these questions can be answered from the discussion provided in the paper.

An interesting feature of the presentation of findings is the use of pronouns. Comments reported from staff perspectives tended to use the collective “we”. If a comment was attributed to a single participant, it was reported as a collective professional responsibility, as in "Professionals need to be equipped…”; “As mental health professionals, we need to be sensitive…” The strategy “institutionalized” staff responses and removed any variation that might have occurred in or between the different groups of participants who could be collectively regarded as “staff”. One might have expected some variations in the perspectives and experiences of sub-groups of staff.

All comments identifiable from their context as students’ quotes are in first person singular; this does allow the voices of individual students to be heard, but there is no attempt to recognize similarities or difference between perspectives of international students and those of local students. For instance, the final theme identified the need for Physical space and Confidentiality, as “participants” noted that the waiting rooms and counseling and therapy rooms did not sufficiently protect the identities of the clients and professionals. Quotes were all identifiably from student participants, but commentary in the Discussion presented these as exclusive concerns for international students. Right to privacy for consultations should have been of equal concern to local students, especially in advocating an inclusive model for mental health care. Although 23.5% of student participants were local students, there is no recognition or inclusion of their perspectives.

The paper focuses on identifying and responding to the mental health needs of international students, but also seeks to promote inclusive mental health values and practices which recognize diversity, “promoting accessible and equitable services for all individuals, irrespective of their cultural background or identity” (p.14, line 11). The incidence of mental health disorders among first year university students, combined with low reported use of mental health services (Osborn, Li, Saunders & Fonagy, 2022) suggests that a more proactive strategy for early engagement of students might benefit all parties. Rather than risking stigmatization of international students as a needy group, a more productive strategies for engaging all students in self-care might go closer to achieving the kind of inclusive mental health support with the potential to serve the needs of a wider spectrum of students that the researchers seek to achieve through their final recommendations.

At present, it is difficult to see that this paper makes a substantial contribution to knowledge in the field, or that it fulfils the claims of the Impact Statement.

Suggested amendments

Adjust the Impact Statement so that it is more in keeping with the Limitations identified in the Conclusion of the paper.

Address the conceptual conflict outlined in Para 2 of this review.

Provide some contextual evidence about the provision of mental health services at the research site. With particular emphasis on the ways in which they might be considered to be inclusive.

Consider providing some demographic details of the participants, although not in a way that might breach their confidentiality.

Provide details of the interview schedules (or at least extracts thereof) .

Explain why and how only some interviews were audio recorded and how data from non-recorded interviews was handled systematically for coding purposes.

Account for the interview data collected from the local students.

References

Letourneau, J. (2016). A Decision-Making Model for Addressing Problematic Behaviors in Counseling Students. In Counseling and Values, October 2016, 61, 206-222.

T. G. Osborn, T.G., Li, S., Saunders, R., and P. Fonagy, P. (2022). University students’ use of mental health services: a systematic review and meta-analysis. International Journal of Mental Health Systems (2022) 16:57 https://doi.org/10.1186/s13033-022-00569-0

---

## [Reviewer Report]

Some literature cited is outdated. There has been an increased body of international students' literature in recent years. While it is common to cite a 20-year-old article which established a theory (e.g. acculturation) or developed an original treatment model, it is preferable to cite recent literature within the last 10 years for specific population studies. Authors should consider updating the reference list to include more recent articles, especially given the dynamics of the international student population.

Further, inadequate data were provided to support the conclusion in the results and discussion sections. Authors should consider expanding the table and providing 1-2 quotes to support each subtheme.

Page 3, line 4, “In recent years..”

Can you please provide global statistics on international students? Also, please use

more recent data (i.e. post 2021 at least to reflect the COVID impact).

Page 3, line 19-26, “Despite the ... Popadiuk and Arthur, 2004)”

Both references are old. Back to 2000-2004, there were not as many international students as there are today.

Page 6. Line 18-20, “Despite...worldwide”

Is there any published statistics on the increase?

Page 6, line 40-42, "First, international students comprise a significant

proportion of the student population in higher education institutions worldwide (Bista, 2018)"

Similarly, is there any statistics?

Page 6, line 60, “Third, research in this area can address the underrepresentation...”

With the fast increase of international student population as stated in the article, there has also been an increased volume of research on international students in recent years. Is there more recent literature or data highlighting the underrepresentation?

Page 7, participant section.

- How did you determine the different groups of stakeholders you wanted to interview?

- What are your reasons for including 4 local students? By “Local”, do you mean US born students?

- How was the first participant in each group identified and recruited?

- How did you determine the sample size?

- As gender is such an important factor in mental health, what was the gender distribution of your participants?

- What is the race/ethnicity of your participants?

- When was your data collected?

Page 8, line 18-23

- How many pilot studies were done? Did you conduct a pilot study among each of your stakeholder groups (international, local, staff, faculties, etc)?

- How did you draw the conclusion that your questions were culturally sensitive? Were you able to conduct your pilot studies on people from different cultural backgrounds?

- Were there any incentives for participation?

Page 8, line 51-52

Please elaborate “audio recordings were made for select interviews”. Why was there selection and how was the selection done? For those unrecorded interviews, how did you transcribe the interview? Given recording is a standard practice in qualitative research.

page 10 - page 11, “In addition, the lack of treatment planning..”

How is lack of treatment planning related to multicultural self-awareness? Is there evidence suggesting that certain cultures are more or less inclined to have structured sessions?

Page 11, line 23, “As one participant observed, ”I felt like my psychologist didn’t understand the cultural nuances"

This quote seems more relevant to the multicultural self-awareness theme.

Page 11 and 15, Focus on Reparative Service

Do you suggest that the individual session = Reparative services approach? In other words, only providing individual session= high focus on Reparative services approach? If so, please clarify and support this definition for the audience who are less familiar with counseling terminology.

Page 10 & Page 15

Authors mentioned a few times about “need for better earlier assessments”. Can you please provide a quote supporting this point?

Page 12 & Page 17

-By faculty, do you specifically mean faculty from the department of psychology? Or do you mean the whole university faculty? By Candidates, do you mean mental health service providers such as counsellors?

- Recent years, some literature have highlighted international students' inclination to talk to professors and other academic faculty instead of counsellors for mental health help. Therefore, it would be helpful if your argument is more clear in defining “faculty engagement” to differentiate.

Page 17

The preference for a local provider reminded the readers about the previous argument on the impact of language proficiency in counseling sessions. It might be helpful to discuss these points together.

Page 18, line 32, “potentially overlooking other relevant..”

Can you please give an example of the other relevant factors?

---

## [Reviewer Report]

Review: Inclusive Mental Health Support for International Students: Unveiling Delivery Components in Higher Education

The study presents a well-structured discussion on the need for further research on the delivery components of inclusive mental health services for international students in higher education in the US. It highlights several strengths, such as the emphasis on multicultural self-awareness among mental health professionals and the significance of faculty involvement in supporting international students. These aspects provide valuable insights and practical recommendations for enhancing mental health support.

However, there are several notable areas where the study could be improved to enhance its impact and validity. While the inclusion of different stakeholders in data collection is commendable, the study primarily focuses on the perspectives of international students. To increase the robustness of the research, it would have been beneficial to explore the viewpoints of other stakeholders, including counseling teams, local students, and faculty staff. Gaining an understanding of different perceptions and potential variations in experiences based on factors such as identity and level of study (undergraduate vs. graduate) would have added depth to the findings.

Moreover, the study mentions the inclusion of counseling staff and faculty members, but it would be advantageous to delve into their understanding of multicultural self-awareness and the specific challenges they encounter in providing counseling to international students. This would provide a more comprehensive picture of the support system and its complexities. See some relevant studies below:

Dovchin, S. (2020). The psychological damages of linguistic racism and international students in Australia. International Journal of Bilingual Education and Bilingualism, 23(7), 804-818.

Dryden, S., & Dovchin, S. (2022). Translingual English discrimination: loss of academic sense of belonging, the hiring order of things, and students from the Global South. Applied Linguistics Review, (0).

In summary, the study contributes valuably to the field of inclusive mental health services for international students in higher education. It offers practical insights and recommendations for universities seeking to improve the quality of mental health support. However, the study’s potential for impact could be further enhanced by broadening the scope to include the perspectives of various stakeholders and by providing more in-depth exploration of the challenges faced by mental health professionals and the practical steps needed to address them. By addressing these aspects, universities can better promote the well-being and academic success of their international student populations.

---

## [Reviewer Report]

12.03.2023

Dear Professor Gary Belkin, PhD

Thank you for giving our manuscript entitled “Inclusive Mental Health Support for International Students: Unveiling Delivery Components in Higher Education” a chance for review. We also thank the Reviewers for reading the manuscript and providing us with con-structive and insightful feedback. We are glad you and the Reviewers have found value in the manuscript and recommended progress with suggestions. We hope and believe that we have worked on the suggestions in a detailed manner and satisfied the concerns. 

We are glad that we have addressed all issues raised by all Reviewers. We now believe that the paper looks more professional, tidy, and scientifically high-quality. We have made revisions within the text and showed the revisions in red color. Also, we provide details of the revisions we made in a separate letter (uploaded as Supplementary Material) titled “Response to Comments”. We quote the Reviewers in the numbered list and provide our answer to each numbered reviewer statement. 

In any case, should you and the reviewers require further revisions, we will be very happy to address them in future cycles. 

Best regards,

Dr. Halis Sakız 

Associate Professor 

Corresponding author

---

## [Reviewer Report]

The author has adequately responded to my suggestions and the paper is publishable now. I have no further suggestions

---

## [Reviewer Report]

I acknowledge that the researchers have addressed all of the reservations raised in my original review and have indicated very clearly how their revisions have been incorporated in the current iteration of their research.

I have no further matters to raise and therefore recommend that the paper be accepted for publication.